# A systematic review and meta-analyses on initiation, adherence and outcomes of antiretroviral therapy in incarcerated people

Terefe G. Fuge *, George Tsourtos, Emma R. Miller

Medicine and Public Health, Flinders University, Adelaide, Australia

* fuge0002@flinders.edu.au

**Data Availability Statement:** All relevant data are within the paper and its Supporting Information files.

## Abstract

### Background

Incarcerated people are at increased risk of human immunodeficiency virus (HIV) infection relative to the general population. Despite a high burden of infection, HIV care use among prison populations is often suboptimal and varies among settings, and little evidence exists explaining the discrepancy. Therefore, this review assessed barriers to optimal use of HIV care cascade in incarcerated people.

### Methods

Quantitative and qualitative studies investigating factors affecting linkage to care, ART (anti-retroviral therapy) initiation, adherence and/or outcomes among inmates were systematically searched across seven databases. Studies published in English language and indexed up to 26 October 2018 were reviewed. We performed a narrative review for both quantitative and qualitative studies, and meta-analyses on selected quantitative studies. All retrieved quantitative studies were assessed for risk of bias. Meta-analyses were conducted using RevMan-5 software and pooled odds ratios were calculated using Mantel-Haenszel statistics with 95% confidence interval at a p<0.05. The review protocol has been published at the International Prospective Register of Systematic Reviews (PROSPERO; Number: CRD42019135502).

### Results

Of forty-two studies included in the narrative review, eight were qualitative studies. Sixteen of the quantitative studies were eligible for meta-analyses. The narrative synthesis revealed structural factors such as: a lack of access to community standard of HIV care, particularly in resource limited countries; loss of privacy; and history of incarceration and re-incarceration as risk factors for poor HIV care use in prison populations. Among social and personal characteristics, lack of social support, stigma, discrimination, substance use, having limited knowledge about, and negative perception towards ART were the main determinants of suboptimal use of care in incarcerated people. In the meta-analyses, lower odds of ART initiation was noticed among inmates with higher baseline CD4 count (CD4 $\geq$500celss/mm$^3$)

**Funding:** The authors received no specific funding for this work.

**Competing interests:** The authors have declared that no competing interests exist.

(OR = 0.37, 95%CI: 0.14–0.97, $I^2$ = 43%), new HIV diagnosis (OR = 0.07, 95%CI: 0.05–0.10, $I^2$ = 68%), and in those who lacked belief in ART safety (OR = 0.32, 95%CI: 0.18–0.56, $I^2$ = 0%) and efficacy (OR = 0.31, 95%CI: 0.17–0.57, $I^2$ = 0%). Non-adherence was high among inmates who lacked social support (OR = 3.36, 95%CI: 2.03–5.56, $I^2$ = 35%), had low self-efficiency score (OR = 2.50, 95%CI: 1.64,-3.80, $I^2$ = 22%) and those with depressive symptoms (OR = 2.02, 95%CI: 1.34–3.02, $I^2$ = 0%). Lower odds of viral suppression was associated with history of incarceration (OR = 0.40, 95%CI: 0.35–0.46, $I^2$ = 0%), re-incarceration (OR = 0.09, 95%CI: 0.06–0.13, $I^2$ = 64%) and male gender (OR = 0.55, 95% CI: 0.42–0.72, $I^2$ = 0%). Higher odds of CD4 count <200cells/mm$^3$ (OR = 2.01, 95%CI: 1.62, 2.50, $I^2$ = 44%) and lower odds of viral suppression (OR = 0.20, 95%CI: 0.17–0.22, $I^2$ = 0%) were observed during prison entry compared to those noticed during release.

## Conclusion

Given the high HIV risk in prison populations and rapid movements of these people between prison and community, correctional facilities have the potential to substantially contribute to the use of HIV treatment as a prevention strategy. Thus, there is an urgent need for reviewing context specific interventions and ensuring standard of HIV care in prisons, particularly in resource limited countries.

## Introduction

Global incarceration rates have increased substantially in the last two decades and there are currently more than 10 million people in prison worldwide [1]. Although there has been a recent decline in the number of new HIV infections in the general population worldwide, the virus is disproportionately affecting people in the prison system. Globally, 3.8% of the incarcerated people are estimated to be HIV infected, which is around five times higher than HIV prevalence in the general population [2]. Risk factors for both incarceration and HIV infection often overlap and include unemployment, poverty, homelessness, and substance use [3].

Despite the dramatic increase in the size of the incarcerated population and associated HIV prevalence, HIV care in correctional facilities is often substandard. While there is evidence that higher rates of linkage to care and subsequent viral suppressions can be achieved in prison populations [4–6], access to community standard of HIV care is often lacking within most prisons, particularly in low-income countries [7–10]. Factors pertaining to insufficient financing, insecurity of food, inadequate health staff and facilities [11–15], as well as lack of integration between community and prison health care systems [6, 16, 17] are considered main structural barriers to HIV care in correctional facilities. Consequently, delayed initiation of ART defined as initiating ART at World Health Organization (WHO) clinical stage III or IV [18], poor adherence and associated clinical complications are highly prevalent in prison populations compared to the general populations [9, 19–21]. Studies have shown that personal and psychosocial factors are also important in the utilization of HIV care among prisoners. Low awareness and negative perceptions of HIV and ART, as well as ongoing substance use are known to contribute to poor utilization of HIV care in prison populations [13, 22–25]. In addition, an increased risk of stress, depression, despair and mental health problems in inmates is associated with high rates of suboptimal ART adherence [24, 26, 27] and virological failure [4]. In some countries, HIV infected people in correctional facilities can face torture, violence, stigma

and discrimination, from both prison staff and other inmates, which could potentially impede care utilization and cause poor treatment adherence and outcomes [10, 12, 13, 28].

Prisoners are an inseparable part of the community regarding HIV transmission. Prisoners interact with the outside society not only after serving their sentences but also during incarceration through contact with prison staff and family visits. Thus, implementation of ART as an HIV prevention strategy [18] in correctional facilities is paramount, given the fact that inmates usually return to the same high risk groups from which they originate, such as people who inject drugs (PWID), sex workers and men who have sex with men (MSM). As access to HIV care for these groups can be challenging in the community, correctional facilities should create an ideal setting to implement such interventions [29].

Utilization of HIV care in correctional facilities varies widely across countries and settings within a country [30]. However, little is known about this variation in relation to promoting best practices and the use of evidence-based interventions. There have been few narrative reviews on prison HIV care with a primary focus on prisons of high-income countries [30–35]. Uthman et al [24] conducted a systematic review and meta-analyses of global studies exclusively on ART adherence among prisoners but the review did not encompass other major care cascade elements such as ART initiation and viral suppression. Iroh et al [6] and Erickson et al [36] also conducted systematic reviews on HIV care cascade in prison systems, but both reviews focused on studies in high-income countries, and the latter was specifically focused on female inmates. Thus, we systematically reviewed global studies investigating one or more of the main components of HIV care cascade (i.e. ART initiation, adherence and/or outcomes) in a prison population, with the intention to identify potential barriers to HIV care use and inform evidence-based intervention strategies for HIV infected people in correctional facilities, and to put forth further research priorities.

## Methods

The reporting of this review was based on the Preferred Reporting Items for Systematic Reviews and Meta-Analyses guidelines (PRISMA) [37] (see S1 Table). The review protocol has been published at the International Prospective Register of Systematic Reviews (PROSPERO; Number: CRD42019135502) [38] (S1 File).

### Eligibility criteria

**Studies.** Both quantitative and qualitative studies were reviewed without restriction based on type of study design and publication date.

**Participants.** All studies in participants with a history of incarceration or currently being incarcerated were considered for review. Studies conducted on specific populations such as certain ethnic groups or populations identified as at high HIV risk or as vulnerable groups (e.g. transgender people, men who have sex with men) were excluded in order to reduce potential confounding, as these groups have been associated with low utilization of care in community and other settings [39, 40].

**Exposures.** Studies exploring structural, social and individual level determinants of HIV care utilization among prisoners were reviewed. More specifically, studies analysing factors related to access to and availability of HIV care; psychosocial factors such as depression, social support, disclosure, stigma and privacy; behavioural factors such as attitudes towards ART; health and medication related factors including comorbidity, immunological or clinical status; incarceration related factors such as number and length of imprisonment; and socioeconomic factors including age, sex, and other characteristics were assessed.

**Comparators.** While no restriction was made based on whether a study has used comparators, non-incarcerated people were considered as a control group when comparisons were made.

**Outcome measures.** Studies reporting one or more of the following outcomes were included in the review: linkage to HIV care, initiation of ART, adherence to and outcomes of ART in terms of change in CD4 count and viral suppression. No restriction was made based on the definition of the outcomes.

## Information sources and search strategy

Systematic searches were carried out on the following databases; Emcare, Medline, PubMed, Scopus, Web of Science, Cinahl and Cochrane Library. The concepts HIV/AIDS, ART and Incarceration were used to construct the search strategy. The search strategy used only terms related to exposure (incarceration) and outcomes. The terms were combined with MEDLINE filter for the concepts under search. The search strategy for MEDLINE was; HIV or AIDS or HIV-AIDS or Acquired Immunodeficiency Syndrome or Human immunodeficiency virus AND antiretroviral* or anti-retroviral* or HAART or ART or anti-hiv AND prison* or incarcerate* or imprison* or inmate* or jail* or detention* or "correctional facilities" or "correctional setting" or "house of correction" or custody or convict* or detain*. The search terms were adapted for use with other bibliographic databases in combination with database-specific filters for the concepts, where these are available. The search strategy was developed with the guidance of a qualified librarian. Bibliographies of the retrieved studies as well as previous meta-analyses were searched for studies that might have been missed by the search strategy and no further studies were identified. While no restriction was made in terms of geographical region and year of publication, due to resource and time restrictions, studies published in English language and indexed up to 26 October 2018 were included in the review. An alert was set for newly indexed articles for each database and no relevant studies were detected post Oct 2018 with the last alert received on 28 March 2020.

## Study selection and risk of bias assessment

Articles were initially screened for relevance with their titles and abstracts. After removal of duplicate and irrelevant articles, a full text review was performed on the retrieved articles based on the predefined protocol [38]. One author (TGF) performed the initial screening and selection of all papers including the quality assessments. Two other authors (GT and ERM) independently conducted the quality assessments (each assessing half of the studies) initially undertaken by the first author (TGF). The quality assessment was conducted using the Effective Public Health Practice Project (EPHPP) Quality Assessment Tool for Quantitative Studies (see S2 File) by considering the following characteristics; representativeness of participants (selection bias), study design, control of potential confounders, validity and reliability of data collection methods and completeness of outcome data (withdrawals and dropouts). Disagreements between the review authors were resolved by discussion, with involvement of a third review author where necessary.

## Data abstraction

Data were extracted using a format adapted from the Cochrane Systematic Review Checklist for Data Collection (see S2 Table). Separate data extraction formats were used for treatment initiation, adherence and outcomes categories. Information in the data extraction form included author, year, geographical location, population, method, measurements, exposures, outcomes and conclusions. Corresponding authors of two primary studies were contacted for

missing information on the number of participants with and without ART initiation and/or non-adherence versus exposure variable of interest.

## Data synthesis

We provided narrative synthesis of the findings across all qualitative and quantitative studies with regard to exposures and outcomes. Due to the variety of outcomes measured and differences in definition of each outcome across studies, our meta-analyses were limited to 16 of 34 quantitative studies included in the narrative review. Meta-analyses were conducted using RevMan-5 software [41] for each outcome when two or more studies assessed the exposure variable. A Fixed Effect Model was employed to pool the outcomes with odds ratios, and calculated 95% confidence intervals. We used a Fixed-Effect Model due to small numbers of studies (n<5) involved in the meta-analyses reporting a particular outcome, which made an estimation of between study variance impossible [42]. In addition, in most of the meta-analyses, a single study had substantially larger sample sizes relative to the other(s) in the model, so that generalization of the findings could not be claimed beyond the included studies [43]. We determined heterogeneity between studies with effect measures using $Chi^2$ test and $I^2$ statistic. We considered an $I^2$ value of 75% as high heterogeneity [44]. Mantel-Haenszel statistics were applied to calculate pooled odds ratios and results were presented in forest plots.

## Results

The search resulted in a total of 2,345 articles. Fig 1 shows the overall screening process and number of studies excluded and retrieved. A total of 2,274 articles were eliminated due to duplication and irrelevance based on title and abstract review. Twenty-nine of the remaining 71 studies were excluded after full text review due to the study not analysing HIV care during incarceration, not reporting at least one of the HIV care cascade elements i.e. linkage to care, ART initiation, adherence or outcomes in terms of CD4 count or viral load or being different reports of the same study. The remaining 42 articles were included in the final review with 16 out of 34 quantitative studies being included in the meta-analyses.

### Study characteristics

The main characteristics of included studies are described in Tables 1–3. Thirty (71%) of the studies were from high-income countries; USA (18), Canada (5) and Europe (7) while the remaining twelve (29%) were from low-and middle-income countries; Asia (5), sub-Saharan Africa (5) and Latin America (2). 41% (17) of the studies were cross-sectional [5, 11, 16, 23, 25, 26, 45–55], 38% (16) cohort (14 retrospective and 2 prospective) [4, 9, 15, 17, 19–21, 56–64] and 19% (8) qualitative [12–14, 28, 65–68] in study design. One study employed mixed methods [22]. Fifteen studies reported on linkage to HIV care or ART initiation or both [5, 9, 11, 12, 22, 23, 45–50, 55, 65, 66] (Table 1), sixteen on adherence [12–15, 17, 20, 25, 26, 28, 45, 51, 52, 55, 56, 67, 68] (Table 2) and twenty on CD4 count or viral load or both [4, 5, 16, 17, 19, 21, 23, 25, 51, 53, 54, 56–64] (Table 3). Nine studies reported more than one element of the HIV care cascade and hence were included in more than one category [5, 12, 17, 23, 25, 45, 51, 55, 56]. Of the 42 articles, 39 were related to incarceration and HIV care, and the remaining three were specific to jail incarceration [46, 48, 57]. Twenty-nine studies investigated HIV care during incarceration [5, 9, 11–15, 19, 22, 23, 25, 26, 28, 45–48, 50–55, 62, 64–68] and seven investigated the impact of history of incarceration and/or the number of incarcerations [16, 17, 20, 21, 49, 57, 63]. Six studies compared HIV care utilization between incarceration trajectories [4, 56, 58–61]; three before and after incarceration [4, 56, 58] and three between incarcerated and re-incarcerated people [59–61]

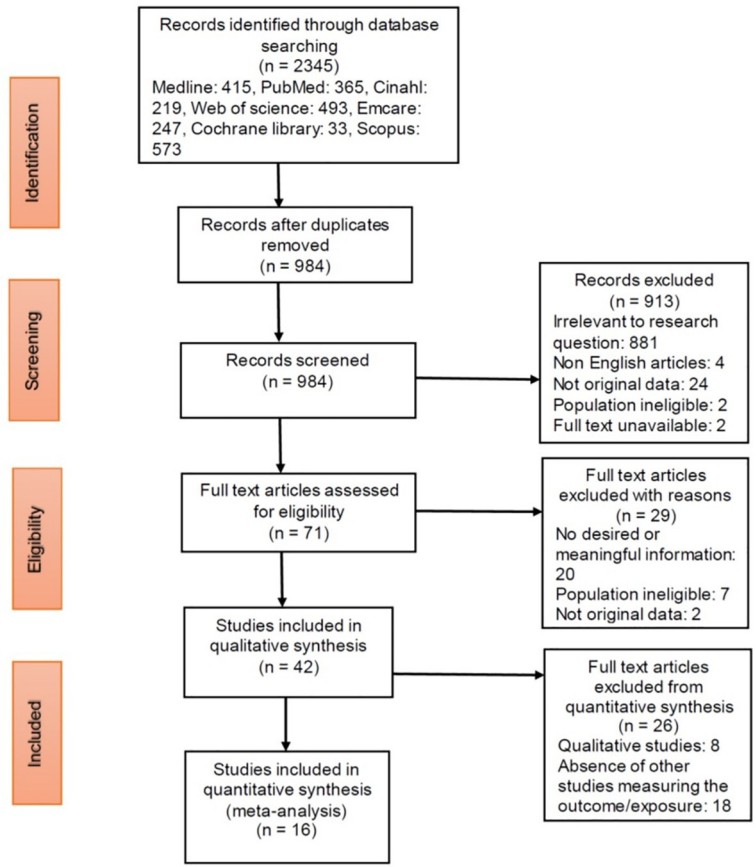

**Fig 1. Study flow diagram.** Study selection process and reasons for exclusion.

## Methodological quality

The majority of studies (74%) were scored as moderate or above performance with regard to minimising selection bias, while 38% scored moderate or above performance in terms of the appropriateness of the study design. Half of the studies (50%) accounted for confounding variables during analyses. Only eleven studies (32%) reported validity of data collection methods with four of these scoring 'strong' on the EPHPP tool. Risk of bias due to drop-out and withdrawal was inapplicable in the majority of studies (85%) mainly due to analyses of retrospective data, but of five studies in which it was applicable, three studies scored moderate or above performance. While two studies had a strong performance in the measurement of the overall methodological quality, eight other studies scored as having moderate methodological quality (see S3 Table).

## Measurements

Definitions of linkage to HIV care and a delay in ART initiation varied across studies. Three studies measured time between diagnosis and linkage to care and/or initiation of ART [5, 47, 50]. Two studies used WHO clinical staging defining a delay as ART initiation at stage III or IV [9, 11]. Other studies simply estimated ART coverage and acceptance (i.e. proportion of prisoners on ART) at a particular period of time [22, 23, 45, 46, 48, 49]. A full description of the definitions is presented in Table 1.

**Table 1. Characteristics of studies investigating linkage to HIV care and initiation of ART.**

| Author | Year | Country | Population | Study Design | Measurement | Findings | Conclusions |
|---|---|---|---|---|---|---|---|
| Mostashari et al [45] | 1998 | USA | 102 ART eligible women prisoners | Cross-sectional | ART acceptance | 75% of the women accepted ART | • Acceptance of first offer of ART associated with completed education lower than high school (OR:3.5, 95%CI:1.2–10.7) and belief in medication safety (OR:4.3, 95%CI:1.3–13.7) • Current acceptance of ART associated with trust in medication efficacy (OR:3.2, 95%CI:1.2–8.6) and safety (OR: 4.3, 95%CI:1.4–12.9) |
| White et al [46] | 2001 | USA | 77 HIV infected jail inmates | Cross-sectional | Percentage of ART initiation | ART initiation 58% in overall; 57% in males; 71% in females;73% in patients with CD4 count $\leq$500cells/mm$^3$; 33% in those with CD4 count $\geq$500cells/mm$^3$ | Lower baseline CD4 count (<500cells/mm$^3$) associated with higher rate of ART initiation (p<0.017) |
| Altice et al [55] | 2001 | USA | 205 HIV infected prisoners eligible for ART | Cross-sectional | Current ART acceptance defined as being prescribed ART at the time of the interview | Acceptance of ART 80% in overall | Mistrust of medication (AOR: 0.3, P<0.001) and trust in physician (AOR: 1.08, P<0.0001) were associated with ART acceptance |
| Perez-Molina et al [49] | 2002 | Spain | 804 non-incarcerated and 104 incarcerated HIV infected individuals | Cross-sectional | Comparison of ART utilization between incarcerated and non-incarcerated people | No descriptive results reported | Incarcerated people utilized ART three times fewer than non-incarcerated people (OR: 2.95, 95% CI: 1.5–6.0) |
| Makombe et al [9] | 2007 | Malawi | 103 HIV infected prisoners | Retrospective cohort (2004–2006) | Estimation of delay in ART initiation | 93% of the prisoners started ART at WHO stage III or IV | Low access to HIV care (challenge: accessing HIV care from outside prison system) |
| Guin [65] | 2009 | India | 10 HIV infected prisoners | Qualitative | Exploration of HIV care service in prison | ——— | Barriers to HIV care: inadequate access to HIV care and support service; protracted structural process to access care from public health facilities |
| Jaffer et al [48] | 2012 | USA | 224 newly identified and 593 known HIV infected jail detainees | Cross-sectional | Percentage of detainees initiating ART | 17% in newly identified; 76% in know HIV patients within 14 days of jail entry | Reasons for not starting ART; short stay (49%) and high CD4 count (39%) in newly identified people; short stay (38%) and being treatment naïve (17%) in known HIV positive people |
| J. Culbert [12] | 2014 | USA | 42 HIV infected male and male-to-female transgendered recently released persons | Qualitative | Men's perception of and experiences with HIV care and ART during incarceration | ———— | Delayed treatment initiation due to lack of status disclosure and medication privacy in fear of stigma, discrimination and violence by prison officers and other inmates |
| Monarca et al [23] | 2015 | Italy | 338 HIV infected prisoners | Cross-sectional | Number of prisoners on ART | 81.4% of the prisoners were on ART | Refusal (69.2%), ongoing medication assessment (23.1%), fear of medication side effects, lack of privacy, religious/ethnic beliefs (7.7%) were reported as reasons for not initiating ART |
| Sgarbi et al [47] | 2015 | Brazil | 34 HIV infected prisoners | Cross-sectional | Number of prisoners initiated ART | 47% of the prisoners started ART within 6-months of diagnosis | No statistical analysis performed |

(*Continued*)

**Table 1.** (Continued)

| Author | Year | Country | Population | Study Design | Measurement | Findings | Conclusions |
|---|---|---|---|---|---|---|---|
| Seth et al [50] | 2015 | USA | 841 newly HIV diagnosed prisoners | Cross-sectional | Linkage to HIV care defined as attendance at first medical appointment after diagnosis | 67.5% linked within any time frame after testing; 37.9% linked within 90 days; 72.3% in older people (≥50years) at any time; 43.8% in younger people (18–29 years) within 90 days of diagnosis | No statistically significant associations observed |
| Bick et al [11] | 2016 | Malaysia | 221 HIV infected male prisoners | Cross-sectional | Prevalence of ART initiation | 34.4% of ART eligible and 22.8% with advanced AIDS not started ART | *Insufficient resource allocation for HIV treatment and care |
| Lucas et al [5] | 2016 | USA | 135 HIV infected prisoners | Cross-sectional | • Linkage to HIV care defined as receiving a CD4 or viral load test within 90 days of HIV diagnosis<br>• ART initiation | • 99% linkage to care and 91% ART initiation within 90 days of diagnosis<br>• Initiation of ART at CD4 count ≥500cells/mm$^3$: 90% in previously diagnosed;50% in newly diagnosed | Longer duration of time (median 28 days) to linkage to care in newly diagnosed cases compared to previously diagnosed cases (median 0 days) (p<0.0001) |
| | | | | | | | * opt-out screening and care approach at prison entry achieved higher rates of linkage to care during incarceration but higher rates of care interruption after release |
| Culbert et al [22] | 2016 | Indonesia | 102 HIV infected prisoners | Mixed method | Number of prisoners starting ART | A quarter of ART eligible prisoners didn't start ART | -ART utilization associated with higher score of attitude towards ART efficacy and safety (OR:1.90, 95% CI: 1.03–1.16) |
| | | | | | | | * Inmates who endorsed the attitude that ART is inefficient, unsafe, and causes adverse effects and stigma were less likely to use the treatment |
| Sprague et al [66] | 2017 | USA | 25 HIV infected women former prisoners | Qualitative | Self-reported experiences in accessing HIV care in prison | ———— | Delay in receiving diagnosis results and structural barriers to see health staff led to delayed treatment initiation |

Study ID (identification), geographical location, population involved, study design and main outcomes of articles included in the analyses of linkage to HIV care and ART initiation

All ART adherence studies measured adherence to dose over varying period of time (days, weeks and months) using different methods (self-report, pharmacy refill, pill count, and electronic monitoring cups). Three studies considered adherence to medication schedule as an alternative measure to dose adherence [25–27]. Table 2 depicts how adherence was defined by the studies. Two studies set optimal adherence at 100% [17, 45] and other two used a threshold of >95% [15, 20]. Three studies defined non-adherence as missing more than three doses or schedules in a week [25–27]. The cut-off for viral suppression also varied greatly among studies, ranging from <40copies/mL [56, 69] to <500copies/mL [17, 21]. Eight studies used <400copies/mL [4, 19, 58–61, 63, 64] and four used <200copies/mL [5, 16, 57, 62]. In this review, we dichotomised the outcomes based on the highest cut off values used in the included studies; adherence <100% as a threshold for non-adherence and viral load <500 copies/mL for viral suppression. Most studies measured immunological outcomes as a change in CD4 count between two points in time (e.g. between entry and release from prison). Table 3 shows definitions of immunological and virological outcomes used by the studies.

**Table 2. Characteristics of studies investigating adherence to antiretroviral therapy.**

| Author | Year | Country | Population | Study Design | Measurement | Findings | Conclusions |
|---|---|---|---|---|---|---|---|
| *Mostashari et al [45] | 1998 | USA | 102 HIV infected female prisoners | Cross-sectional | Adherence defined as taking medication for ≥6 days/week, and not missing any doses per day | Non-adherence in 38% overall | Satisfaction with patient-physician relationship (OR:3.0, 95%CI:1.1–8.5) and seeking emotional supports from others (OR:3.1, 95%CI:1.1–9.4) associated with adherence |
| *Altice et al [55] | 2001 | USA | 164 HIV infected prisoners taking ART | Cross-sectional | Adherence defined as taking 80% or more of the prescribed drugs | Adherence to ART 84% in overall | Composite variable of medication side effects and stopping medication when side effects occur (AOR: 0.09, P = 0.0001), social isolation (AOR: 0.08, P = 0.0005) and complexity of antiretroviral regimen (AOR: 0.33, P = 0.01) were negatively associated with ART adherence |
| Palepu et al [17] | 2004 | Canada | 101 HIV infected people with history of incarceration and 1645without history of incarceration | Retrospective cohort (1997–2002) | Adherence defined as number of days patients received antiretroviral therapy refills divided by number of days of follow-up in the first year after starting therapy | Non-adherent (<100%) in 40% overall; 10% in incarcerated, 3% in non-incarcerated | Non-adherence positively associated with a history of incarceration (OR 2.40, 95% CI: 1.54–3.75) |
| Soto Blanco et al [26] | 2005 | Spain | 177 HIV infected prisoners | Cross-sectional | Non-adherence defined as missing at least 2 doses or schedules in the last 5 days | Non-adherence in 24.3% overall;14% in females; 16% in males; 68% in those not-visited by people from outside; 35% in those reported robbery as a reason for incarceration | Fewer than one family visit in a month (OR:2.21, 95% CI:1.10–4.46), reporting robbery as a reason for imprisonment (OR:2.36, 95% CI:1.01–5.50), difficulty in taking medication (OR:3.64, 95%CI:1.78–7.43), having anxiousness and/or depression (OR:2.43, 95%CI:1.15–5.13) and receiving methadone treatment (OR:2.74, 95% CI:1.08–6.93) were associated with non-adherence |
| Soto Blanco et al [51] | 2005 | Spain | 281 HIV infected prisoners | Cross-sectional | No-adherence defined as more than two doses missed in the last week, or more than 2 days of total non-medication in the last 3 months | Non-adherence in 54.8% overall; 64.6% in prisoners lacking support from officers; 66.7% in prisoners having difficulty in taking medication; 85.7% in prisoners unable to continue medication; 63.6% in mentally ill;83.3% in prisoners lacking support from outside prison | Having difficulty in taking medication (OR:1.94, 95%CI: 1.05–3.57), inability to continue with the medication (lack of self-efficacy) (OR: 5.37, 95%CI: 2.06–13.94), lack of support from outside prison (OR: 3.97, 95%CI: 1.19–13.23) and feeling anxious or depressed (OR: 2.07, 95%CI: 1.18–3.66) were associated with non-adherence |
| White et al [52] | 2006 | USA | 31 HIV infected prisoners | Cross-sectional | Adherence defined as the proportion of prescribed doses taken | No descriptive results reported | Access to ART (correlation coefficient (r) = 0.43, p < 0.05), attitude towards taking ART (r = 0.53, p<0.05), coping scale (r = 0.49, p < 0.05), emotional wellbeing (r = 0.37; p < 0.05) and physical functioning (r = 0.44, p < 0.05) associated with adherence |

(*Continued*)

**Table 2.** (Continued)

| Author | Year | Country | Population | Study Design | Measurement | Findings | Conclusions |
|---|---|---|---|---|---|---|---|
| Ines et al [25] | 2008 | Spain | 50 HIV infected prisoners | Cross-sectional | Non-adherence defined as missing at least 2 medication doses or schedules in the last 5 days | Non-adherence 58% in overall | Predictors of non-adherence: previous injecting drug use (OR: 8.86, 95%CI: 1.52–51.77) |
| | | | | | | | Predictors of adherence: having job in prison (OR: 5.56, 95%CI: 1.12–27.02), absence of HIV-related symptoms (OR: 7.81, 95%CI:1.01–62.5), good or average acceptance of treatment (OR: 10.10, 95%CI: 1.23–83.33) and higher academic background (OR: 5.20, 95%CI: 1.05–26.31) |
| Small et al [67] | 2009 | Canada | 12 HIV positive and IDU male prisoners | Qualitative | Experience with ART in prison | ———— | Barriers to adherence: discrimination leading to discretely taking medication which caused missing of doses; difficulty to obtain medication due to complicated institution health care delivery system particularly during entry; poor relation with physicians; poor quality of health staff |
| Roberson et al [28] | 2009 | USA | 12 HIV infected women prisoners | Qualitative | Factors affecting adherence | ———— | Barriers to adherence: stigma, loss of privacy and long waiting time due to reception of drugs through DOTs; bad treatment by prison officers and other inmates |
| | | | | | | | Facilitators of adherence: tailoring drug taking time with prisoners' routine; support by nurses, friends or officers; using KOP than DOTs; concern for health, a desire to live, and evidence of improved health such as increased CD4 counts |
| Milloy et al [20] | 2011 | Canada | 271 HIV infected IDUs | Retrospective cohort (1996–2008) | Adherence defined as number of days ART dispensed divided by number days that a patient eligible for ART | 61% median level of adherence | Non-adherence (adherence <95%) associated with number of incarceration; 1–2 incarceration events (OR: 1.49, 95% CI: 1.03–2.05); 3–5 events (OR: 2.48; 95% CI: 1.62–3.65); >5 events (OR: 3.11, 95% CI: 1.86–4.95) |
| Paparizos et al [15] | 2013 | Greece | 93 HIV infected prisoners | Longitudinal record review (2001–2011) | -Adherence defined as medication intake according to regimen (>95%) | Regiment or dose non-adherence 56% in overall | Age <40 years associated with non-adherence (p<0.015) |
| Shalihu et al [13] | 2014 | Namibia | 18 HIV infected male prisoners | Qualitative | Identifying barriers to adherence | ———— | Barriers to adherence: lack of medication privacy leading to stigma, lack of social support, low access to food, brutality of officers causing despair, and commodification of ARVs by inmates due to low knowledge about HIV and ART |

(*Continued*)

**Table 2.** (Continued)

| Author | Year | Country | Population | Study Design | Measurement | Findings | Conclusions |
|---|---|---|---|---|---|---|---|
| *J. Culbert [12] | 2014 | USA | 42 HIV infected male and male-to-female transgendered recently released persons | Qualitative | Men's perception of and experiences with HIV care and ART during incarceration | —————— | Barriers to adherence: delayed prescribing, out-of-stock medications, intermittent dosing during lockdowns, poor care and discrimination |
| Seyed Alinaghi et al [14] | 2016 | Iran | 17 HIV infected prisoners | Qualitative | Barriers to ART adherence | —————— | Barriers to adherence: drug addiction, negative drug reactions, bad experiences with staff, psychosocial and nutritional problems, and poor quality of food |
| Subramanian et al [56] | 2016 | Canada | 58 HIV infected prisoners | Retrospective cohort (2007–2011) | Adherence defined as number of months for which ART was dispensed divided by the number of months of follow-up | Mean adherence 57.3% one year before incarceration; 88.7% during incarceration | Adherence during incarceration was significantly higher than adherence before incarceration (p<0.00) |
| Farhoudi et al [68] | 2018 | Iran | 7 HIV infected male prisoners | Qualitative | Barriers and facilitators of adherence | —— | Barriers to adherence: medical factors: drug side-effects, medication interruption, taking methadone maintenance treatment, physical conditions, knowledge about CD4 level and accessibility of complementary medicines; social factors: stigma, patient-physician relationship; psychological factors: depression, anxiety, and disappointment; other factors: lack of education about ART, drug use, forgetfulness and lock ups |

Study ID (identification), geographical location, population involved, study design and main outcomes of articles included in the analyses of adherence to ART

*Studies included in other categories

## HIV care linkage and ART initiation

Rate of linkage to care and ART initiation among inmates varied widely across geographical regions and settings (Table 1). Three studies from high income countries (two from the USA and the other from Italy) reported 75% and above initiation of ART among HIV infected inmates [5, 23, 45]. Lucas et al [5] in the USA identified 99% care linkage among inmates within 90 days of diagnosis. However, three other studies in the same country; one national [50] and two jail studies [46, 48] reported relatively lower rates of linkage to care (66%) and initiation of ART (58% vs 46%), respectively. Similarly, one study in Spain found three times lower utilization of ART by incarcerated people compared to their non-incarcerated counterparts [49].

There were few of published studies on HIV care use in the prisons of low-and middle-income countries, however available studies reported substantial delays in treatment initiation. A national retrospective ART survey in Malawi [9] reported 93% ART initiation among prisoners at WHO stage III or IV. A cross-sectional study in Malaysia reported that fewer than 50% of ART eligible inmates were initiated on treatment, a quarter of whom developed acquired immunodeficiency syndrome (AIDS) [11]. Another cross-sectional study in Brazil

**Table 3. Characteristics of studies investigating outcomes of antiretroviral therapy.**

| Author | Year | Country | Population | Study Design | Measurement | Findings | Conclusions |
|---|---|---|---|---|---|---|---|
| Palepu et al [21] | 2003 | Canada | 234 HIV infected IDUs | Retrospective cohort (1996–2001) | Viral suppression defined as having viral load of <500copies/mL in two consecutive measurements | Viral suppression in 19% in those with history of incarceration; 40% in those without a history of incarceration | Incarceration negatively associated with viral suppression (OR: 0.22, 95% CI: 0.09–0.58) |
| *Palepu et al [17] | 2004 | Canada | 101 HIV infected people with history of incarceration and 1645 without history of incarceration | Retrospective cohort (1997–2002) | Viral suppression defined as having at least two consecutive viral load of <500 copies/mL | Viral suppression in 96% of people without a history of incarceration; 89% in people with a history of incarceration | History of incarceration negatively associated with viral suppression (HR: 0.68, 95% CI: 0.51–0.89); whereas longer time spent in prison was positively associated with viral suppression (HR: 1.06, 95% CI: 1.02–1.10) |
| Springer et al [61] | 2004 | USA | 1866 HIV infected prisoners | Retrospective cohort (1997–2002) | • Viral suppression defined as having viral load of <400 copies/mL<br>• Change in viral load and CD4 count during incarceration | Viral suppression 59% in overall; mean CD4 count increased by 74 cells/mL and the mean viral load decreased by 0.93 log10 copies/mL during incarceration; mean CD4 count decreased by 80 cells/mL, and the mean viral load increased by 1.14 log10 in re-incarcerated | Significant decrease in viral load (p<0.0001) and increase in CD4 count (p<0.0001) during incarceration, whereas significant increase in viral load (p< 0.0001) and decrease in CD4 count (p< 0.0001) at re-incarceration |
| Stephenson et al [60] | 2005 | USA | 15 re-incarcerated and 30 incarcerated HIV infected males | Retrospective cohort (1997–1999) | • Viral suppression defined as having viral load of <400 copies/mL<br>• Change in CD4 count over the follow period | Viral suppression at the beginning 53% in re-incarcerated; 50% in non-re-incarcerated; 20% in re-incarcerated at the end of two and half years follow up; 47% in non-re-incarcerated; mean CD4 count at the beginning 224 cells/mm$^3$ in re-incarcerated; 446 cells/mm$^3$ in non-re-incarcerated; 157 cells/mm$^3$ in re-incarcerated at the end of the follow up; 560 cells/mm$^3$ in non-re-incarcerated | Re-incarceration associated with poor immunological (p<0.003) and virological (OR: 8.29, 95% CI:1.78, 38.69) outcomes |
| *Soto Blanco et al [51] | 2005 | Spain | 281 HIV infected prisoners | Cross-sectional | • Viral suppression defined as having viral load of <log$_{10}$ 1.6 copies/mL<br>• CD4 count | Viral suppression in 60.5% overall; mean viral load, log10 4.69 copies/ml; mean CD4 count, 381cells/mm$^3$; mean viral load 4.68 in adherent; 5.12 in non-adherent; mean CD4 count, 390.55cell/mm$^3$ in adherent; 373.53cells/mm$^3$ in non-adherent | No individual factors associated with viral suppression and mean CD4 count |
| *Ines et al [25] | 2008 | Spain | 50 HIV infected prisoners | Cross-sectional | • Virological failure defined as having viral load of >50 copies/mL<br>• Change in CD4 count after treatment | Viral suppression 46% in overall; change in CD4 count within 6-months of ART 119.71 ± 29.75 in overall; mean HIV-RNA levels 1.68 ± 0.26 log10 copies/mL in adherent patients; 1.33 ± 0.33 log10 copies/mL in non-adherent; change in CD4 count 188.21 ± 55.83 cells/mm$^3$ in adherent and 70.10 ± 28.84 cells/mm$^3$ in non-adherent patients | Adherence significantly associated with undetectable viral load (p< 0.004) and increase in CD4 count (p<0.048) |

(*Continued*)

**Table 3.** (*Continued*)

| Author | Year | Country | Population | Study Design | Measurement | Findings | Conclusions |
|---|---|---|---|---|---|---|---|
| Westergaard et al [63] | 2011 | USA | 437 HIV infected IDUs | Prospective cohort (1998–2009) | • Virological failure defined as having viral load of >400 copies/mL<br>• CD4 count | Virological failure 53.3% in those incarceration reported; 24.8% in those no incarceration reported; CD4 count of <200 cells/mm$^3$ 24% in never incarcerated; 26.5% in at least once incarcerated; viral load of >10,000 copies/mL 37.4% in never incarcerated; 43.6% in at least once incarcerated | Brief incarceration (7–30 days) associated with virological failure (both at 400 and 10,00 cepies/mL cut offs) (OR: 7.7, 95%CI: 3.0–19). |
| Davies and Karstaedt [19] | 2012 | South Africa | 148 HIV infected prisoners | Retrospective cohort (2004–2008) | • Viral suppression defined as having viral load of <400 copies/mL<br>• Change in median CD4 count over the ART period | Viral suppression in 73% overall; median CD4 count 122 cells/mm$^3$ during ART initiation; 356 cells/mm$^3$ after 96 weeks of treatment | No statistical analysis performed |
| Meyer et al [4] | 2014 | USA | 882 HIV infected prisoners | Retrospective cohort (2005–2012) | • Viral suppression defined as having viral load of < 400 copies/mL<br>• Change in CD4 count between entry and release | Viral suppression 29.8% in overall during entry; 70% during release; 68.% in men; 79.1% in women; 63.6% in psychiatric patients; 72.1% in non-psychiatric patients; mean increase in CD4 count 98 cells/μl during incarceration; mean decrease in viral load 1.12 log10 during incarceration | Viral suppression correlated with female sex (OR:1.81, 95% CI:1.26–2.59) and low psychiatric problem (OR: 1.50, 95% C: 1.12–1.99); significant increase in CD4 count (P < 0.001) and decrease in vital load (P< 0.001) during incarceration |
| Meyer et al [59] | 2014 | USA | 497 HIV infected prisoners | Retrospective cohort (2005–2012) | • Viral suppression defined as having viral load of <400 copies/mL<br>• Change in viral load and CD4 count between release and re-incarceration | Viral suppression 70% in overall before release; 52% in recidivists before release; 31% in recidivists on re-incarceration; mean loss of CD4 count 50.8 cells/mm$^3$ between release and re-incarceration; mean viral rebound $0.4\log_{10}$ between release and re-incarceration | Recidivism negatively associated with viral suppression (p<0.0001); increase in age (OR:1.04, 95% CI:1.01–1.07) and having higher level of medical or psychiatric comorbidity (OR:1.16, 95%CI:1.03–1.30) associated with viral suppression during re-incarceration |
| *Monarca et al [23] | 2015 | Italy | 338 HIV infected prisoners | Cross-sectional | • Viral suppression defined as having viral load of <50 copies/mL<br>• CD4 Count | Viral suppression 73.5% in overall; >200 cells/mm$^3$ 90.6% in overall | No statistical analysis performed |
| Meyer et al [58] | 2015 | USA | 1,089 HIV infected prisoners | Retrospective cohort (2005–2012) | • Viral suppression defined as having viral load of <400 copies per/mL<br>• Change in CD4 count during incarceration | Average viral suppression, 32.7% at entry; 70.6% during release; 80% in females; 68.7% in males; mean CD4 count, 344.5 cells/mm$^3$ at entry; 449.5 cells/mm$^3$ during release | Significantly more viral suppression rate in women than men during pre-release (p<0.002) |
| Chan et al [54] | 2015 | England | 74 HIV infected prisoners | Cross-sectional | Viral suppression defined as having viral load of <40 copies/mL | Viral suppression 68% in overall | No statistical analysis performed |

(*Continued*)

**Table 3.** (Continued)

| Author | Year | Country | Population | Study Design | Measurement | Findings | Conclusions |
|---|---|---|---|---|---|---|---|
| *Lucas et al [5] | 2016 | USA | 83 HIV infected prisoners | Cross-sectional | • Viral suppression defined as having viral load of <200 copies/mL<br>• Change in CD4 count during incarceration | Viral suppression at late assessment 88% in overall; median increase in CD4 count 160cells/mm$^3$ in overall; viral suppression 43% at initial assessment in newly diagnosed; 25% in previously diagnosed; 86% at late assessment in newly diagnosed; 81% in previously diagnosed inmates | Significant change in viral suppression (p<0.0001) and CD4 count (p<0.0001) during incarceration |
| *Subramanian et al [56] | 2016 | Canada | 58 HIV infected prisoners | Retrospective cohort (2007–2011) | • Viral suppression defined as having viral load of <40copies/mL<br>• Change in CD4 count during incarceration | Viral suppression 50% in overall at prison entry;77.8% at exit; CD4 count of <200cells/mm$^3$ 57.1% at entry;61.9% at exit | CD4 count significantly improved during incarceration (p<0.02) |
| Nasrullah et al [16] | 2016 | USA | 443 HIV infected people with history of incarceration and 8077 without history of incarceration | Cross-sectional | Viral suppression defined as having viral load of <200copies/mL | Viral suppression 55.8% in incarcerated; 74.2% in non-incarcerated people | Recently incarcerated persons are significantly less likely to achieve viral suppression (OR:0.90, 95%CI:0.86, 0.95) |
| Telisinghe et al [64] | 2016 | South Africa | 404HIV infected prisoners | Retrospective cohort (2007–2009) | Viral suppression defined as having viral load of <400copies/mL | Viral suppression 94.7% in ART naïve prisoners at 6$^{th}$ month;92.5% at 12$^{th}$ month; 72.1% in ART experienced prisoners at 6$^{th}$ month | *Provision of onsite ART service yielded high percentage of viral suppression |
| Eastment et al [57] | 2017 | USA | 202HIV infected people with history of jail booking and 6788 without history of jail booking | Retrospective cohort (2014) | Viral suppression defined as having viral load of < 200 copies/ml or no viral load report | Proportion of CD4 count <200cells/mm$^3$, 25% in people with a history of jail booking; 7% in people without a history of jail booking; Viral suppression 62% in people with a history of incarceration (one year after release);79% in non-incarcerated people | Incarceration associated with lower CD4 count (p<0.001) |
| Mpawa et al [53] | 2017 | Malawi | 262HIV infected prisoners | Cross-sectional | Viral suppression defined as having viral load of <40 copies/mL. | Viral suppression in 95% overall | No patient characteristics associated with viral suppression |
| Dos Santos Bet et al [62] | 2018 | Brazil | 25HIV infected prisoners | Prospective cohort (2013–2014) | • Viral suppression defined as having viral load of < 200 copies/mL | Viral suppression 46% in overall | No statistical analysis performed |

Study ID (identification), geographical location, population involved, study design and main outcomes of articles included in the analyses of ART outcomes (change in CD4 count and viral load)

*Studies included in other categories

[47] reported less than 50% initiation of ART among HIV infected prisoners with unknown clinical status within 6-months of diagnosis.

Different personal and structural factors have been identified as factors affecting ART initiation among HIV infected prisoners. Lucas et al [5] and Jaffer et al [48] found longer time of linkage to care in those who were diagnosed during prison entry compared with those diagnosed before prison entry. The same authors [5, 48] and White et al [46] noted higher rates of ART initiation among inmates with lower baseline CD4 count (<500cells/mm$^3$). ART acceptance in HIV infected inmates was also influenced by their attitudes towards the medication.

Mostashari et al [45] and Altice et al [55] found a higher rate of ART acceptance among inmates who perceived ART as safe to take and efficient in improving their health. A similar finding was obtained by Culbert et al [22] who found that ART utilization among inmates was associated with more positive attitudes towards medication safety and efficacy.

While there is yet to be strong evidence for specific factors contributing to delayed presentation for care among HIV infected inmates in low-and middle-income countries, lack of access to standard of care has been proposed as a major barrier to ART initiation. Makombe et al [9] in Malawi reported limited access to HIV care in prisons as HIV infected inmates were forced to receive ART services from public health facilities. Bick et al [11] reported minimal resource allocation for prison HIV care in Malaysia compared to care provided in the community, which resulted in delayed treatment initiation and frequent care interruptions among inmates. A qualitative study in India [65] supported these findings by exploring protracted structural processes involved in accessing care from public health facilities which prevented HIV infected inmates from starting ART. Barriers to ART initiation among inmates appeared to possess different account in the context of prisons of high income countries. Qualitative studies from the USA [12, 66] described the importance of institutional and social barriers to care despite the presence of a standard of HIV care being provided in the correctional facilities. Prisoners dissuaded from disclosing their HIV status being afraid of perceived stigma and discrimination against, as well as anticipating a violent response by officers and other fellow inmates, which rendered them initiate treatment delayed.

## Adherence to ART

Studies investigating the impact of incarceration history on ART adherence identified higher odds of non-adherence in people with a history of incarceration than those without a history of incarceration [17, 20]. One study in the USA on the other hand compared adherence at prison entry and exit, finding a significant increase in the level of optimal adherence during incarceration (57% vs 89%) [56]. Six other studies from different countries estimated prevalence of non-adherence among inmates during incarceration [15, 25–27, 45, 52, 70], and the overall prevalence ranged from 24% [26] to 58% [25] (Table 2).

Structural, social, and behavioural factors were found to affect inmates' adherence to ART. Among structural factors, Soto Blanco et al [26] identified higher rates of non-adherence in individuals who were incarcerated due to robbery offences, presumably due to shorter sentences. White et al [52] found more non-adherence in those who reported inconvenience in accessing care from the prison health care system. Qualitative studies [12–14, 28, 67] similarly explained a number of institutional-related factors to affect adherence among inmates including lack of privacy during medication pick-ups and use, difficulty in accessing care, and insufficiency and/or poor quality of food.

Social support within prison and from the outside community was associated with inmates' adherence to ART. Mostashari et al [45] and Altice et al [55] found optimal adherence in prisoners who were able to seek emotional support from others, and those who established good relationships with their care provider. Qualitative studies emphasized the importance of inmate-health care provider relationships in enhancing optimal adherence [14, 28, 67, 68]. Other studies [28, 67] also showed that inmates were more likely to use ART when health care providers were found to be caring and sympathetic towards their clients. Higher ART adherence was observed among inmates who reported 'cooperative' prison officers [27], and among those who were able to engage in jobs in prison [25]. This was concordant with what was described by qualitative studies [12–14, 28, 67, 68] that alienation of inmates using ART by prison officers and other inmates resulted in suboptimal adherence. Soto Blanco et al [26] and

Blanco et al [27] on the other hand identified higher prevalences of adherence in inmates who were capable of receiving regular visits from people from outside prison.

Behavioural factors and attitudes towards ART were reported to influence adherence. Ines et al [25] found that the inmate's belief in ART efficacy and safety had an effect on ART adherence. A study by White et al [70] corroborated the association between the inmate's belief in ART efficacy and ART adherence that those who believed that ART would help them live longer were more likely to be adherent. Two other studies [26, 27] documented higher likelihood of non-adherence in prisoners with difficulty of taking medication and those who could not consistently follow their medication schedule (commonly reported as having low self-efficacy).

Other behaviour- and awareness-related factors were suggested to influence ART adherence among prisoners: history of injecting drug use, medication refusal, and unintended use of ARV drugs as a result of having little knowledge about HIV and the health importance of ART were described as risk factors for non-adherence [13, 15, 25, 68]. Difference in adherence was also observed among inmates based on age and academic background. Paparizos et al [15] showed a high probability of poor adherence among inmates aged younger than 40 years compared to older prisoners. Ines et al [25] reported higher adherence among those with a higher academic background.

Factors related to individual health appeared to affect inmates' adherence to ART. In their two consecutive studies, Soto Blanco et al [26] and Blanco et al [27] identified a strong association between depression and suboptimal adherence among prisoners. White et al [70] supported this association using different scales of adherence measurement (i.e. medication admission record and pill count). Qualitative studies [14, 68] also highlighted the impact of depression on inmates' adherence as depressed prisoners lacked motivation to use ART due to being hopeless for recovery. Ines et al [25] on the other hand demonstrated that the presence of any non-specific symptoms of illness increased the probability of non-adherence. This was concordant with findings by White et al [70] and Farhoudi et al [68], which showed a relationship between inmates' emotional and physical wellbeing and ART adherence.

## ART outcomes

Five studies investigated the impact of incarceration history on viral suppression [16, 17, 21, 57, 63], with two of these simultaneously analysing change in CD4 count over the course of treatment [57, 63]. In all cases, a statistically significant increase in viral suppression and CD4 count was recorded in people without a history of incarceration compared to those with a history of incarceration (Table 3). Four studies from high-income countries analysed changes in viral suppression and CD4 count during incarceration [4, 5, 56, 58]. All studies showed an increase in both treatment outcomes during the course of incarceration. In the studies that investigated the association between re-incarceration and ART outcomes [4, 60, 61], a statistically significant increase in viral load and decrease in CD4 count was observed among people with episodes of re-incarceration. Eight studies reported the overall rate of viral suppression during incarceration [19, 23, 25, 27, 54, 62, 64, 69], which ranged from 46% in Spain [25] and Brazil [62] to 95% in Malawi [69]. Four of these studies reported on CD4 count; two measuring mean and median CD4 count (381cells/mm$^3$ and 356 cells/mm$^3$, respectively) [19, 27], and the other two reporting change in CD4 count within 6-months of ART commencement (119.71 ± 29.75 cell/mm$^3$) and the percentage of inmates with CD4 count >200 cells/mm$^3$ (91%) [23, 25].

There was inconsistency among rarely available published studies about specific factors affecting viral suppression and CD4 count among HIV infected inmates. Ines et al [25] identified a higher level of viral suppression and an increase in CD4 count in adherent inmates

compared to non-adherent inmates. However, Blanco et al [27] reported no statistical association between adherence and viral suppression or CD4 count, although there were lower viral load and higher CD4 count in adherent prisoners than non-adherent ones. Meyer et al [4] in the USA found a negative association between psychiatric disorder and viral suppression among inmates. Two consecutive studies by these same authors also identified a correlation between female sex and viral suppression during incarceration [4, 58]. Whilst male and female inmates had comparable viral suppression at prison entry, females possessed significantly higher odds of achieving viral suppression during incarceration. In contrast, Mpawa et al [69] in Malawi found no association between viral suppression and inmate characteristics.

### Meta-analyses of factors affecting ART initiation, adherence and outcomes

Meta-analyses for each outcome was employed when at least two studies assessed the exposure variable. The Fixed Effect Model was applied as the number of studies involved in the meta-analyses of a particular outcome was low, and considerable difference in size existed between the studies [42, 43]. The effect of incarceration history on CD4 count was not analysed because of a high level of heterogeneity between the studies reporting the outcome ($I^2$ = 96%). Mantel-Haenszel statistics was applied to calculate pooled odds ratio and the results are presented using forest plot as shown in Figs 2A–5B.

Sixteen studies involving 22,190 people were included in the meta-analyses to determine factors associated with initiation, adherence and outcomes of ART among prisoners. Lower odds of ART initiation was noticed among inmates with higher baseline CD4 count (CD4 $\geq$500celss/mm$^3$) (Fig 2A; OR = 0.37, 95%CI: 0.14–0.97, $I^2$ = 43%), new HIV diagnosis (Fig 2B; OR = 0.07, 95%CI: 0.05–0.10, $I^2$ = 68%), and in those who lacked confidence in ART safety (Fig 2C; OR = 0.32, 95%CI: 0.18–0.56, $I^2$ = 0%) and efficacy (Fig 2D; OR = 0.31, 95%CI: 0.17–0.57, $I^2$ = 0%).

Non-adherence was high among inmates who lacked social support (Fig 3A; OR = 3.36, 95%CI: 2.03–5.56, $I^2$ = 35%), had low self-efficiency score (Fig 3B; OR = 2.50, 95%CI: 1.64,-3.80, $I^2$ = 22%) and those with depressive symptoms (Fig 3C; OR = 2.02, 95%CI: 1.34–3.02, $I^2$ = 0%).

Lower odds of viral suppression were associated with a history of incarceration (Fig 4A; OR = 0.40, 95%CI: 0.35–0.46, $I^2$ = 0%), re-incarceration (Fig 4B; OR = 0.09, 95%CI: 0.06–0.13, $I^2$ = 64%) and male gender (Fig 4C; OR = 0.55, 95%CI: 0.42–0.72, $I^2$ = 0%).

Higher odds of CD4 count <200cells/mm$^3$ (Fig 5A; OR = 2.01, 95%CI: 1.62, 2.50, $I^2$ = 44%) and lower odds of viral suppression (Fig 5B; OR = 0.20, 95%CI: 0.17–0.22, $I^2$ = 0%) were observed during prison entry compared to those noticed during release. A study by Lucas et al [5] was removed from the analyses of viral suppression during prison entry and exit to avoid severe heterogeneity (Fig 5B).

## Discussion

The review offered evidence that despite the prisoners' acceptance of, and compliance with ART, issues related to accessibility and availability of standard of HIV care remained a challenge. Although there existed variation at individual and facility levels [50], HIV infected inmates were generally capable of timely initiating ART in prison settings where an acceptable standard of care was available [5, 23, 45]. From the limited available studies of prisons in the low-and middle-income countries, there remained particular challenges in accessing the standard of care available to the surrounding community, and this resulted in delayed treatment initiation and associated health complications [9, 11]. Nevertheless, there was evidence that inmates could respond well to ART in these settings when an appropriate standard of care was

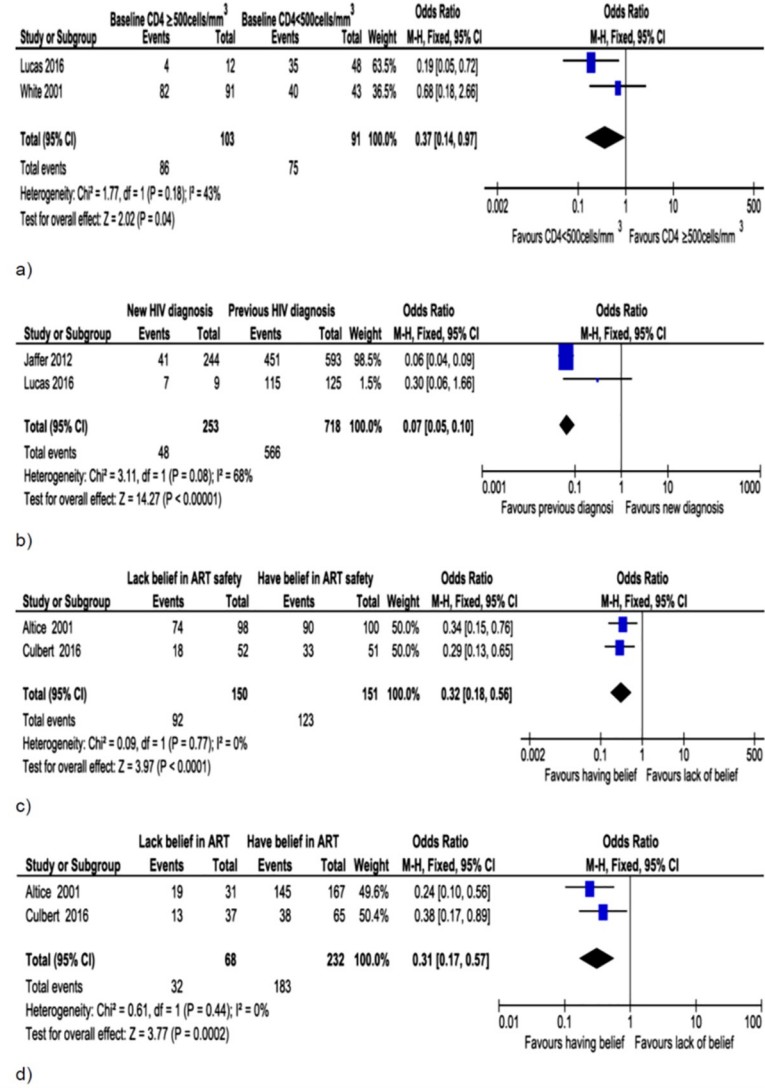

**Fig 2.** Forest plot of associations between ART initiation and baseline CD4 count (a), time of HIV diagnosis (b), belief in ART safety (c) and efficacy (d). Prisoners with higher baseline CD4 count (CD4 ≥500cells/mm³) and new HIV diagnosis, and those who lacked belief in ART safety and efficacy were less likely to initiate ART.

provided [64, 69]. Lower rate of ART initiation was also observed in jail settings which hold people serving short-term sentences, often for less than one year [46, 48], compared to prisons or long-term correctional facilities, possibly as a result of the transient nature of the incarcerated population.

Due to the bureaucracies commonly existing in prison systems, HIV infected inmates often faced challenges in navigating and using ART even in prison settings where the standard of care was available. Suboptimal treatment provided by health care providers, as well as stigma and discrimination arising amongst fellow inmates and prison security, contributed to delayed linkage to care and inadequate adherence to ART [12, 13, 28, 65–67]. In contrast, as supported by the meta-analyses results, inmates who were able to receive support either from people in prison (prison-officers, health staff and other inmates) or people external to prison, such as family and friends, demonstrated good adherence [25–27, 45]. Confidentiality around the use

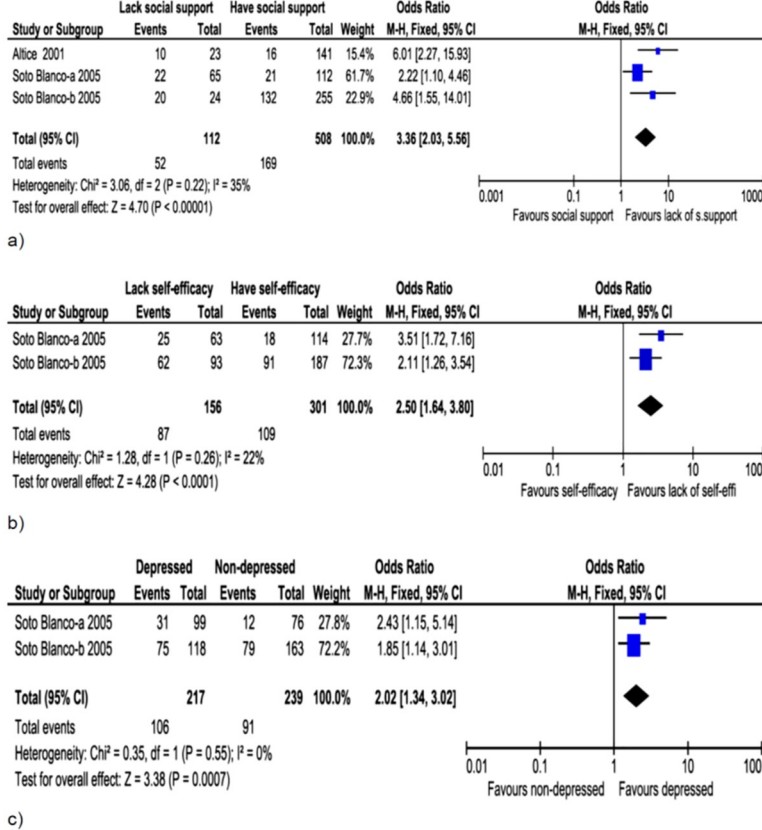

**Fig 3.** Forest plot of associations between non-adherence and social support (a), self-efficacy (b) and depression (c). Inmates who lacked social support, were unable to consistently use ART (or lacked self-efficacy) and those with experience of depression were less likely to be adherent to ART.

of ART seemed to be difficult to maintain in prison, particularly in settings where prisoners were required to form a line [28] or shuttled in a group to external health facilities to access care [13]. Therefore, strategies ensuring medication privacy and the availability of social supports are highly needed in prison systems beyond offering of a high standard of care through adaptation of the Seek, Test, and Treat (STT) strategy [71] which involves identification and offering of ART to all HIV infected individuals, to the unique needs of prison settings.

It was found that the inmate's perception plaid a crucial role in the initiation of and adherence to ART. HIV-infected prisoners may feel healthy during the early phases of their infection and hesitate to initiate ART. This was shown by the current meta-analyses in which inmates with high CD4 count and those newly diagnosed for HIV were more reluctant to start ART [5, 48]. However, care providers at times preferred to prescribe medication for those who had lower CD4 counts [46]. Several studies also reported the same problem in the general populations as people at the asymptomatic stage often hesitate to decide to start ART due to the perception that they are not sick enough to warrant treatment [72, 73]. Prisoners' perception of the safety and efficacy of ART was another important factor affecting their initiation and proper use of ART. Inmates appeared to accept and adhere well to ART when they perceived that it improves health without causing harm [22, 25, 45, 70]. Adherence also occurred when they believed that they possessed the self-efficacy to consistently use the medication for life [26, 27]. It seems that novel information dissemination strategies including peer education

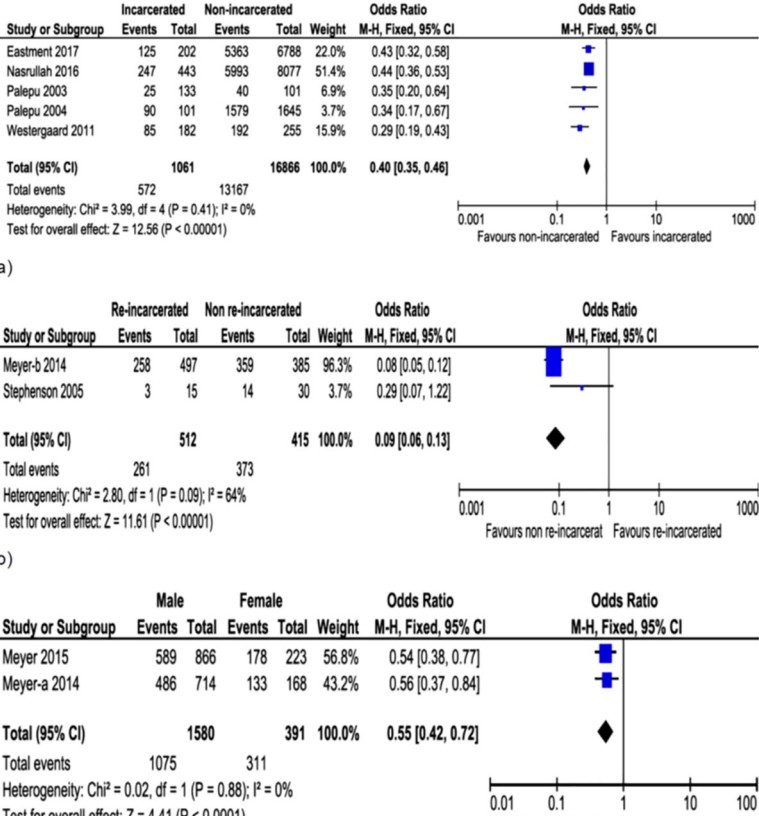

**Fig 4.** Forest plot of associations between viral suppression and incarceration (a), re-incarceration (b) and gender (c). Incarcerated people were at higher risk of viral non-suppression compared to unincarcerated people but had lower risk than re-incarcerated people. Higher odds of viral suppression in females than males at exit from prison.

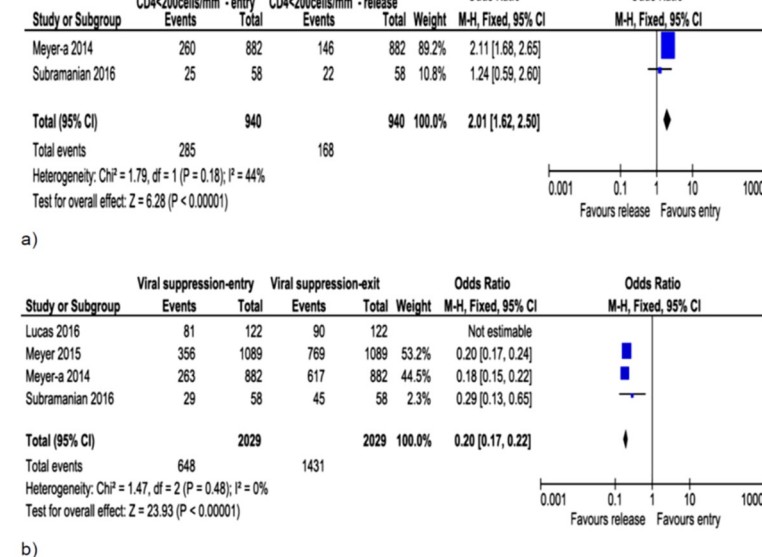

**Fig 5.** Forest plot of differences in CD4 count (a) and viral suppression (b) at prison entry and exit. Higher odds of low CD4 count (CD4 <200cells/mm$^3$) and viral non-suppression at entry than at exit from prison.

and engagement of socially concordant navigators [74, 75] are highly required at prison settings to enhance inmates' awareness of the health benefits of early ART initiation and mechanisms to manage adverse effects of ARV drugs. Effective implementation of international guidelines to initiate all HIV infected individuals on ART regardless of their clinical background could also help minimise treatment delays [76].

Other personal and behavioural characteristics were also found to influence inmates' adherence to ART and subsequent treatment outcomes. Prisoners in some settings possessed limited knowledge about HIV and the importance of ART and so developed indifference to use the medication [13, 15]. Further, males, younger inmates, those with a lower educational background, and those with a history of injecting drug use were at high risk of suboptimal adherence and poor treatment outcomes [4, 15, 25, 58]. Given the high prevalence of these characteristics in incarcerated people [58, 77], group specific HIV care intervention strategies including provision of adequate educational information about HIV and the importance of ART are highly recommended.

Mental health problems were another important determinant of ART adherence and outcomes in prison populations. Due to the high prevalence of depression both in HIV infection [78] and in incarcerated people [79], prisoners infected with HIV were at increased risk of bearing the burden of psychiatric problems, which often caused difficulty in maintaining ART adherence [14, 26, 27, 68, 70], and led to poor treatment outcomes [4]. Integration of HIV care and treatment of medically diagnosed depression is therefore likely to be very important.

Although the level of ART adherence and outcomes varied greatly among studies (range; 42%-89% for adherence and 46%-95% for viral suppression), significant improvements were noted in general during incarceration [4, 5, 56, 58]. The variation might partly be attributed to the difference in overall quality of care provided across settings but might also be influenced by differences in study design and case definition. However, the overall improvements in ART adherence and outcomes during incarceration noted in our systematic review may suggest effectiveness of ART service in correctional facilities.

History of incarceration was associated with poor ART adherence [17, 20] and outcomes [16, 17, 21, 57, 63]. A number of factors might have contributed to this including poor quality of care and other psychosocial as well as structural barriers to care during incarceration. Linkage to community health care system also remains a challenge for maintaining the HIV care continuum among people discharged from the criminal justice system [6]. Moreover, re-incarcerated people were more likely to face viral rebound and immunological suppression than incarcerated people mainly due to care interruptions during their previous release [59–61]. This suggests a need for novel intervention strategies to ensure continuity of care during and after incarceration through integration of prison and community health care systems.

This review is subject to the following limitations. The majority of studies analysing determinants of ART initiation, adherence and outcomes were in high-income countries which made international extrapolation of the findings difficult. Causality between variables could not be claimed as the analyses were mostly made based on retrospective data. We were unable to ascertain determinants of HIV care use in prison settings in low-and middle-income countries as almost all the included studies were simple descriptive studies lacking explicit analyses of the potential factors. The definitions of HIV care cascade elements (i.e. linkage to care, ART initiation, adherence and outcomes) differed among studies, which might have led to over- or under-estimation of the effects. The certainty of the evidence could only be established with low-level of quality as all of the included studies were non-randomized observational studies; only 29% of the studies had a score of 'moderately high' or above in the overall quality assessment and there was inconsistency of effects between the studies (for some of the outcomes) and imprecision of the results as most of the studies were small studies with few events [80].

Studies published in languages other than English were excluded from the review due to resource and time constraints and this might have increased the potential for reporting bias. Also, there could be missed studies as screening was performed by a single reviewer [81]. A funnel plot for the detection of publication bias was not reported due to the small number of studies (n<10) [82] included in the meta-analyses of each exposure variable.

## Conclusion

This systematic review demonstrated that prisoners respond well to ART when they are able to access a standard of care. In addition to the imperative to provide best quality care on an individual level, this finding is of critical public health importance regarding using treatment as an infection prevention strategy as people in prison are at high risk of acquiring HIV infection and transmitting to others in the outside community after their release. Thus, ensuring access to a standard of HIV care at prison settings is paramount. Each prison environment appeared to possess unique circumstances which potentially influence HIV care use, therefore prompting a need to design context specific interventions focusing on structural, social and behavioural aspects. Further research on specific determinants of HIV care use in correctional facilities with a particular focus on low-income countries is highly recommended. Additionally, standardized measures for HIV care cascade outcomes including linkage to care, adherence and viral suppression are crucial.

## Supporting information

**S1 Table. Systematic review reporting checklist.** The preferred reporting items for systematic reviews and meta-analyses guidelines (PRISMA) 2009 checklist for reporting a systematic review.
(DOC)

**S2 Table. Data extraction form.** Data extraction form adapted from Cochrane review format for data extraction.
(DOCX)

**S3 Table. Quality assessment results.** Quality assessment results for quantitative studies included in the final review using EPHPP Tool.
(DOCX)

**S1 File. Systematic review protocol.** A review protocol registered in international prospective register of systematic reviews (PROSPERO).
(PDF)

**S2 File. Study quality assessment tool.** Effective public health practice project (EPHPP) quality assessment tool for quantitative Studies.
(PDF)

## Acknowledgments

We would like to acknowledge authors of the primary studies for the provision of additional information.

## Author Contributions

**Conceptualization:** Terefe G. Fuge.

**Formal analysis:** Terefe G. Fuge.

**Writing – original draft:** Terefe G. Fuge.

**Writing – review & editing:** Terefe G. Fuge, George Tsourtos, Emma R. Miller.

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
