## [Decision Letter · Decision Letter 0]

16 Jan 2020

PONE-D-19-29373

A systematic review and meta-analyses on initiation, adherence and outcomes of antiretroviral therapy in incarcerated people

PLOS ONE

Dear Mr Fuge,

Thank you for submitting your manuscript to PLOS ONE. After careful consideration, we feel that it has merit but does not fully meet PLOS ONE’s publication criteria as it currently stands. Therefore, we invite you to submit a revised version of the manuscript that addresses the points raised during the review process.

We would appreciate receiving your revised manuscript by Mar 01 2020 11:59PM. To enhance the reproducibility of your results, we recommend that if applicable you deposit your laboratory protocols in protocols.io, where a protocol can be assigned its own identifier (DOI) such that it can be cited independently in the future. For instructions see: http://journals.plos.org/plosone/s/submission-guidelines#loc-laboratory-protocols

We look forward to receiving your revised manuscript.

Kind regards,

Stéphanie Baggio

Academic Editor

PLOS ONE

Journal Requirements:

2. Please provide any updates you might have since the original search was performed in October 2018, or please provide the rational for ending your search at that time.

Additional Editor Comments (if provided):

In addition to the reviewers’ comments, I also have a couple of concerns that need to be addressed.

1. The manuscript needs English editing. For example, there ae typos in the abstract (p. 2, line 34, a space before the comma) and grammatical errors (p. 2, lines 36-37, a same sentence uses past one time and present two times; p. 2 line 43, use of “than” inadequately). These are only examples, the whole manuscript needs proofreading (also for awkward labels, such as developed/developing world”).

2. Did two independent authors should screen and select all papers?

3. Did you also search the reference lists of previous meta-analyses on the topic?

4. Please note that a low heterogeneity is not an indication to use fixed-effect models. This is a common mistake. The model should be chosen according to the kind of studies sampled. Please read for example this paper:

https://www.meta-analysis-workshops.com/download/common-mistakes2.pdf

5. Please explain how and why 16 out of 42 studies were selected for meta-analysis. This should also be added in the study flow diagram.

6. Please add the information on risk of bias for each study. A supplementary table with all information used in the meta-analysis should be added to allow replication.

7. p. 32 first paragraph of the subsection “meta-analysis […]” repeats what is already mentioned in the Methods section.

8. Better quality figures are needed, the forest plots are almost not readable. Please also add title to each figure (forest plot) so the reader will be able to understand.

9. Conclusion: I am not sure that HIV treatment should be considered primarily as a “prevention strategy” (abstract p. 2 and conclusion p. 38). Of course it can contribute to protect the whole society, but the primary focus is to provide care (and equivalence of care) to prisoners.

10. What about a conclusion on the methods of future studies? It seems that it would be useful to standardized outcomes for linkage to care, adherence, and viral suppression.

Reviewers' comments:

Reviewer's Responses to Questions

**Comments to the Author**

1. Is the manuscript technically sound, and do the data support the conclusions?

Reviewer #1: Partly

Reviewer #2: Yes

2. Has the statistical analysis been performed appropriately and rigorously? 

Reviewer #1: I Don't Know

Reviewer #2: Yes

3. Have the authors made all data underlying the findings in their manuscript fully available?

Reviewer #1: Yes

Reviewer #2: Yes

4. Is the manuscript presented in an intelligible fashion and written in standard English?

Reviewer #1: Yes

Reviewer #2: Yes

5. Review Comments to the Author

Reviewer #1: I red this paper with great interest and think it is potentially contain interesting information, but i think the paper suffer from serious methodological concerns.

1. First concern that is critical is that when conducting a systematic review, as well as meta-analysis the uniform study design is very important. Given the "Prisma" statement which the authors stated that they have used it, the study design is important as noted in Objectives, introduction section of Prisma checklist item 4 "Provide an explicit statement of questions being addressed with reference to participants, interventions, comparisons, outcomes, and study design (PICOS)" and Eligibility criteria and Data items, Methods section, items 6 and 11 "Specify study characteristics (e.g., PICOS, length of follow-up) and report.." and " List and define all variables for which data were sought (e.g., PICOS, ...)", respectively. Thus, selecting studies with different study design could lead to high heterogeneity in the results and misleading meta-analysis. Thus, different study designs addressing the same question yielded varying results, with differences in about half of all examples. This way could include the risk of presenting uncertain results without knowing for sure.

I think this study more fit to the scoping review category even not systematic review, and suggest to authors to read the following article:

Munn Z, Peters MDJ, Stern C, Tufanaru C, McArthur A, Aromataris E. Systematic review or scoping review? Guidance for authors when choosing between a systematic or scoping review approach. BMC Med Res Methodol. 2018 Nov 19;18(1):143. doi: 10.1186/s12874-018-0611-x. PubMed PMID: 30453902; PubMed Central PMCID:PMC6245623.

2. Why author selected only papers published in English? Though the English language is generally perceived to be the universal language of science. However, the exclusive reliance on English-language studies may not represent all of the evidence. Excluding languages other than English (LOE) may introduce a language bias and lead to erroneous conclusions.

3. In the Prisma flow diagram you selected 16 papers for meta-analysis! these papers have different study designs, so how can do meta-analysis? why your Forrest plots only contain only for 2 to approximately for papers?

4. most of selected papers are cross-sectional design, why the authors calculated ORs instead of RR?

5. Why authors didn't performed funnel plots? The funnel plot is a method to assess the potential role of publication bias (Harbord et al 2006). It assumes small studies are more likely to be susceptible to publication bias than large ones, and it is this difference which is detectable. If a researcher completes a large randomized trial they are likely to want to see it published even if the result is negative because of the effort involved. For small trials, however, the situation may be different. If publication bias does exist, it is most likely to be due to small negative trials not being published.

Reviewer #2: Terefe Gone Fuge and colleagues did an excellent literature review assessing factors affecting linkage to care, ART initiation, adherence and outcomes among people living in detention with HIV in developed and developing countries. This information is crucial to improve access to care for people living with HIV in detention, as the prison population carries a high burden of the disease. This work is also very useful, in order to demonstrate further research priorities, especially in developing countries, where there are data gaps.

I think this manuscript needs minor revision to be considered for publication.

Objectives are clearly stated. The literature review design is appropriate. The narrative synthesis and the statistical analysis have been performed appropriately. The results are clearly presented. The conclusions are consistent and appropriate, based on the data presented.

Two elements need improvement/clarification

1) Relationship between ART initiation and CD4 count.

Odds of ART initiation was explored according to CD4 count. However, indications for ART initiation have varied over the last 2 decades. In particular, since 2015, with the START and Temprano studies, all patients with HIV must be treated regardless of CD4 count. Therefore, the indications at the time of each study must be specified (patients with an indication for treatment). The interpretation of an association of ART not initiated among people with elevated CD4 counts in the studies, for example, of White and colleagues (2001) or Jaffer and colleagues (2012), is totally different compared with studies including patients recruited within the last 5 years (table 1, pages 28 and 33, fig.2a). To be specified also in the discussion/limitations section.

2) Quality of studies

The analysis of the quality (risk of bias) of quantitative studies was evaluated. It revealed that three-quarters of the studies were scored as moderate or above performance with regard to minimising selection bias. Can you estimate the possible impact on results?

6. PLOS authors have the option to publish the peer review history of their article (what does this mean?). If published, this will include your full peer review and any attached files.

Reviewer #1: No

Reviewer #2: No

---

## [Author Response · Author response to Decision Letter 0]

18 Feb 2020

Journal Requirements:

Response: The manuscript has been amended to satisfy all the journal requirements. 

2. Please provide any updates you might have since the original search was performed in October 2018, or please provide the rational for ending your search at that time.

Response: An alert has been set for newly indexed articles for each database and to date no relevant studies have been detected post Oct 2018.

3. PLOS requires an ORCID iD for the corresponding author in Editorial Manager on papers submitted after December 6th, 2016. Please ensure that you have an ORCID iD and that it is validated in Editorial Manager.

Response: An ORCID iD has been created for the corresponding author and validated in Editorial Manager.

Response to editor’s questions 

1. The manuscript needs English editing. For example, there ae typos in the abstract (p. 2, line 34, a space before the comma) and grammatical errors (p. 2, lines 36-37, a same sentence uses past one time and present two times; p. 2 line 43, use of “than” inadequately). These are only examples, the whole manuscript needs proofreading (also for awkward labels, such as developed/developing world”). 

Response: the manuscript has been proofread with editing made where necessary.

2. Did two independent authors should screen and select all papers?

Response: One author (TGF) performed initial screening and selection of all papers including the quality assessments whereas two other authors (GT and ERM) collaboratively carried out and approved the quality assessments (each assessing half of the studies) initially undertaken by the first author (TGF). This now has been clarified in the paper with the inclusion of the following statement (p. 7 line 151):

“One author (TGF) performed the initial screening and selection of all papers including the quality assessments. Two other authors (GT and ERM) independently conducted the quality assessments (each assessing half of the studies) initially undertaken by the first author (TGF).”

3. Did you also search the reference lists of previous meta-analyses on the topic?

Response: Bibliographies of the retrieved studies as well as previous meta-analyses were searched for studies that might have been missed by the search strategy and no further studies were identified. This information has been included in the manuscript (p. 6 line 143). 

4. Please note that a low heterogeneity is not an indication to use fixed-effect models. This is a common mistake. The model should be chosen according to the kind of studies sampled. Please read for example this paper:

Response: It is true that low heterogeneity doesn’t guarantee the existence of a common effect between studies to use fixed-effect models. We used a Fixed-Effect Model due to small numbers of studies (n<5) involved in the meta-analyses reporting a particular outcome, which made an estimation of between study variance impossible [1]. In addition, in most of the meta-analyses, a single study had substantially larger sample sizes relative to the other(s) in the model, so that generalization of the findings could not be claimed beyond the included studies [2]. The methods section of the manuscript has been modified to include this information (p. 8 line 175). 

5. Please explain how and why 16 out of 42 studies were selected for meta-analysis. This should also be added in the study flow diagram.

Response: Due to the variety of outcomes analysed and differences in definition of each of the outcomes across studies, our meta-analyses were limited to 16 out of 34 quantitative studies (8 of the total 42 articles included in the systematic review being qualitative studies) included in the narrative review. Each outcome was considered for meta-analyses when two or more studies assessed the exposure variable. This information has been added in the modified study flow diagram.

6. Please add the information on risk of bias for each study. A supplementary table with all information used in the meta-analysis should be added to allow replication. 

Response: A supplementary table (S5 Table) containing information on risk of bias for each quantitative study has been added (p. 49 line 794).

7. p. 32 first paragraph of the subsection “meta-analysis […]” repeats what is already mentioned in the Methods section.

Response: this is to make the review comply with the International Prospective Register of Systematic Review (PROSPERO) guidelines which recommends a brief description of how meta-analyses were carried out in the Results section, even though it is included in the Methods section. However, we have removed some of the less important details in the Results section to avoid repetition (p. 32 line 394). 

8. Better quality figures are needed, the forest plots are almost not readable. Please also add title to each figure (forest plot) so the reader will be able to understand.

Response: the figures were prepared using PACE (a tool proved by PLOS for preparation of figures) to make them accord with PLOS figure guidelines. I have understood from PLOS publication assistants that the system compresses the quality of figures uploaded during the process of review PDF creation, but the full quality figures can be downloaded from the PDF. Furthermore, I noticed that the figure guidelines do not support inclusion of Captions as part of the figures rather suggest insertion in the text of the manuscript after the section in which the figure are first cited. However, the label of the figures have been matched with the name of the files uploaded at submission to help clarify for the reader. 

9. Conclusion: I am not sure that HIV treatment should be considered primarily as a “prevention strategy” (abstract p. 2 and conclusion p. 38). Of course it can contribute to protect the whole society, but the primary focus is to provide care (and equivalence of care) to prisoners.

Response: In addition to its effect in the prevention of AIDS and no-AIDS related comorbidities, early initiation of ART is believed to reduce new HIV infections by considerably suppressing viral concentration in the infected individuals. This is assumed to have even more importance in the case of the most at risk populations (MARPS) such as prisoners as high coverage of effective ART has strongly been associated with a decline in the risk of HIV acquisition in the society at large. 

10. What about a conclusion on the methods of future studies? It seems that it would be useful to standardized outcomes for linkage to care, adherence, and viral suppression.

Response: Agreed; an amendment has been made accordingly (p.38 line 542). 

Response to reviewers’ questions

Reviewer #1: 

Q1a. I red this paper with great interest and think it is potentially contain interesting information, but i think the paper suffer from serious methodological concerns. First concern that is critical is that when conducting a systematic review, as well as meta-analysis the uniform study design is very important. Given the "Prisma" statement which the authors stated that they have used it, the study design is important as noted in Objectives, introduction section of Prisma checklist item 4 "Provide an explicit statement of questions being addressed with reference to participants, interventions, comparisons, outcomes, and study design (PICOS)" and Eligibility criteria and Data items, Methods section, items 6 and 11 "Specify study characteristics (e.g., PICOS, length of follow-up) and report.." and " List and define all variables for which data were sought (e.g., PICOS, ...)", respectively. Thus, selecting studies with different study design could lead to high heterogeneity in the results and misleading meta-analysis. Thus, different study designs addressing the same question yielded varying results, with differences in about half of all examples. This way could include the risk of presenting uncertain results without knowing for sure.

I think this study more fit to the scoping review category even not systematic review, and suggest to authors to read the following article: Munn Z, Peters MDJ, Stern C, Tufanaru C, McArthur A, Aromataris E. Systematic review or scoping review? Guidance for authors when choosing between a systematic or scoping review approach. BMC Med Res Methodol. 2018 Nov 19;18(1):143. doi: 10.1186/s12874-018-0611-x. PubMed PMID: 30453902; PubMed Central PMCID: PMC6245623.

Response: We thank the reviewer for their comments and we acknowledge the importance of homogeneity of studies for meta-analyses from a clinical and methodological point of view. However, our decision to conduct a meta-analyses was not based purely on statistical heterogeneity (i.e. difference in design and risk of bias). This is because conceptually (clinically) similar studies may be combined in order to increase the likelihood of precision although in this case, findings should be interpreted with caution [2]. A large number of published studies may be extracted for a systematic review that are based on research conducted at a range of settings and circumstances, it is practically difficult to obtain studies which are identical to each other in every aspect. Thus, we selected studies for meta-analyses when at least two studies addressed particular exposure (intervention) and outcome variables in prison populations (i.e. conceptually similar studies), although they might have been conducted at different settings using different designs [3]. 

Q2. Why author selected only papers published in English? Though the English language is generally perceived to be the universal language of science. However, the exclusive reliance on English-language studies may not represent all of the evidence. Excluding languages other than English (LOE) may introduce a language bias and lead to erroneous conclusions.

Response: appreciating the reviewer’s concern about the issue, we were unable to include studies published in languages other than English because of resource and time constraints. We have added an explanation for this in the text (p. 38 line 527).

Q3. In the Prisma flow diagram you selected 16 papers for meta-analysis! these papers have different study designs, so how can do meta-analysis? why your Forrest plots only contain only for 2 to approximately for papers?

Response: the issue of methodological heterogeneity is addressed above. Our meta-analyses of a particular outcome involved a small number of studies primarily due to differences in the type of outcomes measured and definitions given to each outcome across studies. Further, studies were considered for meta-analyses when two or more studies investigated particular exposure and outcome variables.

Q4. most of selected papers are cross-sectional design, why the authors calculated ORs instead of RR?

 Response: ORs are more applicable for cross-sectional designs than RR, which is mostly applicable in follow up studies such as a cohort studies where incidence rates are calculated. ORs are assumed to be the preferred effect size for the meta-analyses of binary data regardless of the study design of the studies [2] in addition to the fact that it appeared to give values that are similar for all the studies included in the meta-analyses [4]. 

Q5. Why authors didn't performed funnel plots? The funnel plot is a method to assess the potential role of publication bias (Harbord et al 2006). It assumes small studies are more likely to be susceptible to publication bias than large ones, and it is this difference which is detectable. If a researcher completes a large randomized trial they are likely to want to see it published even if the result is negative because of the effort involved. For small trials, however, the situation may be different. If publication bias does exist, it is most likely to be due to small negative trials not being published. 

Response: although a funnel plot is a useful tool to test the presence of possible publication bias in meta-analyses, presenting it for a small number of studies (<10) is likely to produce unreliable results [5]. We raised this as a limitation of the review (p. 38 line 529) in discussing that we didn’t report this due to the small number of studies included in the meta-analyses of each exposure variable. 

Reviewer #2:

 Q1. Relationship between ART initiation and CD4 count.

Odds of ART initiation was explored according to CD4 count. However, indications for ART initiation have varied over the last 2 decades. In particular, since 2015, with the START and Temprano studies, all patients with HIV must be treated regardless of CD4 count. Therefore, the indications at the time of each study must be specified (patients with an indication for treatment). The interpretation of an association of ART not initiated among people with elevated CD4 counts in the studies, for example, of White and colleagues (2001) or Jaffer and colleagues (2012), is totally different compared with studies including patients recruited within the last 5 years (table 1, pages 28 and 33, fig.2a). To be specified also in the discussion/limitations section.

Response: We share the reviewer’s concern and the effect of CD4 count on ART initiation appeared to hold two implications in the current review; first, variation in the use of baseline CD4 count as ART eligibility criteria between settings, and secondly, patients with higher baseline CD4 count being often hesitant to start ART presumably because of the perception that they are literally healthy even if the guidelines allowed them to initiate ART. We have tried to take account of these concerns in order to rule out the first case by combining studies in the meta-analyses. For instance, the two studies by Lucas et al (2016) and White et al (2001), studies were combined to assess the effect of baseline CD4 count on ART initiation, applied similar CD4 eligibility criteria although conducted at different times. Similarly, the other two studies by Lucas et al., 2016 and Jaffer et al., 2012 (combined to analyse the impact of being newly diagnosed for HIV on ART initiation) used the same CD4 criteria. However, results of other studies included in the narrative review were interpreted with great caution and this has been discussed as a limitation of the review (p. 37 line 519). 

Q2. Quality of studies

The analysis of the quality (risk of bias) of quantitative studies was evaluated. It revealed that three-quarters of the studies were scored as moderate or above performance with regard to minimising selection bias. Can you estimate the possible impact on results?

Response: The certainty of evidence in the current systematic review can only be established with low-level of quality as all of the included studies are non-randomized observational studies despite the fact that three-quarters of the studies were scored as moderate or above performance with regard to minimising selection bias. The higher score of the included studies in minimizing selection bias was possibly attributed to the fact that the majority of studies were facility based and largely analysed retrospective data, which substantially reduced non-response rates, but the representativeness of the results still cannot be assured, as participants were not randomly selected from comprehensive lists of individuals in the target populations. The certainty of the evidence is also subject to a potential downgrading as only 29% of the studies had a score of ‘moderately high’ or above in the overall quality assessment. Moreover, there might have been bias due to other factors which could have decreased the quality of the evidence including the inconsistency of effects between the studies (heterogeneity), imprecision (as most of the studies were small studies including few events) and the effect of publication bias, which we could not determine because of a small number of studies involved in the meta-analysis of a particular outcome [4]. We explained this in the manuscript by including the following information (p. 37 line 521)

“In addition, the certainty of the evidence could only be established with low-level of quality as all of the included studies were non-randomized observational studies. Further, only 29% of the studies had a score of ‘moderately high’ or above in the overall quality assessment, there was inconsistency of effects between the studies (for some of the outcomes), and imprecision of the results as most of the studies were small studies with few events.”

References 

1. Borenstein M, Hedges L, Rothstein H. Meta-Analysis: Fixed effect vs. random effects. 2007.

2. Tufanaru C, Munn Z, Stephenson M, Aromataris E. Fixed or random effects meta-analysis? Common methodological issues in systematic reviews of effectiveness. International Journal of Evidence-Based Healthcare. 2015;13(3):196-207.

3. Rueda S, Mitra S, Chen S, et al. Examining the associations between HIV related stigma and health outcomes in people living with HIV/AIDS: a series of meta-analyses. BMJ Open 2016; 6:e011453. doi:10.1136/bmjopen-2016- 011453.

4. Schünemann HJ, Higgins JPT, Vist GE, Glasziou P, Akl EA, Skoetz N, Guyatt GH. Chapter 14: Completing ‘Summary of findings’ tables and grading the certainty of the evidence. In: Higgins JPT, Thomas J, Chandler J, Cumpston M, Li T, Page MJ, Welch VA (editors). Cochrane Handbook for Systematic Reviews of Interventions version 6.0 (updated July 2019). Cochrane, 2019. Available from www.training.cochrane.org/handbook.

5. Sedgwick P, Marston L. How to read a funnel plot in a meta-analysis. BMJ: British Medical Journal. 2015; 351:h4718.

---

## [Decision Letter · Decision Letter 1]

28 Apr 2020

PONE-D-19-29373R1

A systematic review and meta-analyses on initiation, adherence and outcomes of antiretroviral therapy in incarcerated people

PLOS ONE

Dear Mr Fuge,

Thank you for submitting your manuscript to PLOS ONE. After careful consideration, we feel that it has merit but does not fully meet PLOS ONE’s publication criteria as it currently stands. Therefore, we invite you to submit a revised version of the manuscript that addresses the points raised during the review process.

We would appreciate receiving your revised manuscript by Jun 12 2020 11:59PM. To enhance the reproducibility of your results, we recommend that if applicable you deposit your laboratory protocols in protocols.io, where a protocol can be assigned its own identifier (DOI) such that it can be cited independently in the future. For instructions see: http://journals.plos.org/plosone/s/submission-guidelines#loc-laboratory-protocols

We look forward to receiving your revised manuscript.

Kind regards,

Stéphanie Baggio

Academic Editor

PLOS ONE

Additional Editor Comments (if provided):

I am very sorry for the late reply on your paper. As you imagine, we have been very busy with the current pandemic and related clinical care and research projects.

Besides, I was able to get the feed-back of one reviewer.

Here are my comments:

Journal requirements

2. Please update the methods section by explaining that no relevant studies were detected post oct. 2018 (with the date of the last alert).

Editor

2. Single screening is less efficient than double screening. It influences the number of studies missed. Please at least add a limitation in the discussion section.

Waffenschmidt, S., Knelangen, M., Sieben, W., Bühn, S., & Pieper, D. (2019). Single screening versus conventional double screening for study selection in systematic reviews: a methodological systematic review. BMC medical research methodology, 19(1), 132.

Fig 1. Please add n for the last box on the right (“absence of other studies measuring the outcome/exposure).

Reviewers' comments:

Reviewer's Responses to Questions

**Comments to the Author**

1. If the authors have adequately addressed your comments raised in a previous round of review and you feel that this manuscript is now acceptable for publication, you may indicate that here to bypass the “Comments to the Author” section, enter your conflict of interest statement in the “Confidential to Editor” section, and submit your "Accept" recommendation.

Reviewer #2: All comments have been addressed

2. Is the manuscript technically sound, and do the data support the conclusions?

Reviewer #2: Yes

3. Has the statistical analysis been performed appropriately and rigorously? 

Reviewer #2: Yes

4. Have the authors made all data underlying the findings in their manuscript fully available?

Reviewer #2: Yes

5. Is the manuscript presented in an intelligible fashion and written in standard English?

Reviewer #2: Yes

6. Review Comments to the Author

Reviewer #2: At the time of the first assessment, I had pointed out that 2 elements needed improvement / clarification. The explanations and adaptations made in the article are appropriate, concerning a) Relationship between ART initiation and CD4 count, and b) Quality of studies.

7. PLOS authors have the option to publish the peer review history of their article (what does this mean?). If published, this will include your full peer review and any attached files.

Reviewer #2: No

---

## [Author Response · Author response to Decision Letter 1]

2 May 2020

Journal Requirements:

1. Please update the methods section by explaining that no relevant studies were detected post Oct. 2018 (with the date of the last alert)

Response: The methods section has been updated to explain that no relevant studies were detected post Oct 2018 with the last alert received on 28 March 2020 (p. 7 line 148). 

Response to editor’s questions: 

1. Single screening is less efficient than double screening. It influences the number of studies missed. Please at least add a limitation in the discussion section. Waffenschmidt, S., Knelangen, M., Sieben, W., Bühn, S., & Pieper, D. (2019). Single screening versus conventional double screening for study selection in systematic reviews: a methodological systematic review. BMC medical research methodology, 19(1), 132.

Response: a limitation has been added to indicate that there could be missed studies as screening was performed by a single reviewer (p. 42 line 611).

2. Fig 1. Please add n for the last box on the right (“absence of other studies measuring the outcome/exposure).

Response: number of studies to which there were no other studies measuring the same outcome/exposure has been specified.

---

## [Editor Report · Decision Letter 2]

5 May 2020

A systematic review and meta-analyses on initiation, adherence and outcomes of antiretroviral therapy in incarcerated people

PONE-D-19-29373R2

Dear Dr. Fuge,

We are pleased to inform you that your manuscript has been judged scientifically suitable for publication and will be formally accepted for publication once it complies with all outstanding technical requirements.

With kind regards,

Stéphanie Baggio

Academic Editor

PLOS ONE
---

## [Editor Report · Acceptance letter]

7 May 2020

PONE-D-19-29373R2 

A systematic review and meta-analyses on initiation, adherence and outcomes of antiretroviral therapy in incarcerated people 

Dear Dr. Fuge:

I am pleased to inform you that your manuscript has been deemed suitable for publication in PLOS ONE. Congratulations! Your manuscript is now with our production department. 

With kind regards,

on behalf of

Dr. Stéphanie Baggio 

Academic Editor

PLOS ONE